# Synthesis of Aluminum Phosphate-Coated Halloysite Nanotubes: Effects on Morphological, Mechanical, and Rheological Properties of PEO/PBAT Blends

**DOI:** 10.3390/nano12172896

**Published:** 2022-08-23

**Authors:** Kui Wang, Xuefei Jin, Xiuhong He, Weijiang Huang, Qin Tian, Qiuping Fu, Wei Yan

**Affiliations:** School of Chemistry and Materials Engineering, Guiyang University, Guiyang 550005, China

**Keywords:** polymer blends, inorganic particles, mechanical property, rheology

## Abstract

Polymer blending has been widely used to fabricate polymeric films in the last decade due to its superior properties to a single component. In this study, an aluminum phosphate-coated halloysite nanotube (HNTs@AlPO_4_) was fabricated using a one-pot heterogeneous precipitation method, organically modified HNTs@AlPO_4_ (o-HNTs@AlPO_4_) was used to improve the performance of polyethylene oxide/poly(butylene adipate-co-terephthalate) (PEO/PBAT) blends, and the mechanical and rheological properties of the PEO/PBAT/o-HNTs@AlPO_4_ films were systematically discussed. According to our results, there is an optimal addition for adequate AlPO_4_ nanoparticle dispersion and coating on the surface of HNTs, and organic modification could improve the interfacial compatibility of HNTs@AlPO_4_ and the polymeric matrix. Moreover, o-HNTs@AlPO_4_ may serve as a compatibilizer between PEO and PBAT, and PEO/PBAT/o-HNTs@AlPO_4_ films have better mechanical and rheological properties than the PEO/PBAT blends without the o-HNTs@AlPO_4_ component.

## 1. Introduction

Polymeric films have been widely used in heavy and light industries, dentistry, petrochemical, and several other fields due to their unique optical, mechanical, electromagnetic, and gas-sensing properties [1,2,3,4]. However, the pollution of refractory polymeric films has become a pressing issue that must be addressed immediately. Recently, because of their sustainable, renewable, and biodegradable properties, research on biopolymer films has become a hotspot for addressing the environmental pollution caused by polymeric films [5,6]. Moreover, blending two polymers has become a strategy for fabricating polymeric films with better properties than single component films [7,8,9].

Polyethylene oxide (PEO) can be used to synthesize water-soluble films and textile pulp due to its low toxicity, bioadhesion, good water solubility, and easy processing and molding. In addition, PEO can form complexes with several organic low molecular compounds and inorganic electrolytes and can be used as a polymer solid electrolyte film for lithium-ion transfer [10,11,12,13]. However, pure PEO is a flexible chain and uncharged semicrystalline polymer with poor mechanical and rheological properties. Therefore, blending PEO with other polymers is one method to solve this problem. Poly(butylene adipate-co-terephthalate) (PBAT) has been frequently blended with other polymers because of its biodegradability, rubber-like behavior, deformability, and toughness [14,15,16,17,18,19,20]. PBAT can improve the comprehensive performance of the blend system. However, polymer compatibility is a major challenge in polymer blending, and further interfacial compatibility improvement is necessary. Currently, one of the main strategies to minimize the interfacial tension of polymeric blends is to use a compatibilizer.

Inorganic particles have been widely used as fillers to improve the interfacial compatibility of polymeric blends due to their strong interfacial interactions and good dispersion [21,22,23,24,25]. Among the various inorganic particles, nanoparticles are the most widely used because of their large specific surface area and unique nanoscale effects. In previous studies, numerous nanoparticles such as carbon-based nanoparticles, montmorillonite, halloysite nanotubes (HNTs), SiO_2_, and Al_2_O_3_ have been introduced into polymeric blends [26,27,28,29,30]. Dil et al. achieved a balance between toughness and stiffness by adding spherical nanoSiO_2_ particles into poly(lactic acid) (PLA)/PBAT blends, and discussed the effect of the interfacial assembly of nanosilica on the morphology, rheology, and mechanical properties of the polymeric blends [18]. Urquijo et al. blended PLA/PBAT with different carbon nanotube (CNT) contents and demonstrated that CNTs selectively located in the PBAT phase induced the coalescence of PBAT spherical particles [19]. Our previous work demonstrated that sepiolite (Sep) improved the toughness of PP/ABS, enhanced both the mechanical and thermal properties of PP/PA6, and functions as a nucleating agent and compatibilizer between polymers [31,32].

In this study, we designed and fabricated aluminum phosphate-coated halloysite nanotube (HNTs@AlPO_4_) composite first time by a simple one-pot heterogeneous precipitation strategy. The influence of the AlPO_4_ contents on the morphology of HNTs@AlPO_4_ was discussed to adjusting the dispersion of AlPO_4_ nanoparticles on the HNTs. Furthermore, in order to improve the poor mechanical and rheological properties of PEO, PBAT was blended into PEO given its good physicochemical properties as described above, and a silane coupling agent was used to modify HNTs@AlPO_4_ to improve the compatibility with the polymeric matrix, and its potential applications as fillers in PEO/PBAT were discussed. Scanning electron microscopy (SEM) was used to investigate the morphology of the HNTs@AlPO_4_ and PEO/PBAT/HNTs@AlPO_4_ films. X-ray powder diffraction (XRD) was used to determine the crystallinity of HNTs@AlPO_4_. Fourier transform infrared spectroscopy (FTIR) was used to investigate the modification of HNTs@AlPO_4_ by the silane coupling agent, CG-570. Thermogravimetric/derivative thermogravimetric analysis (TG/DTG) was used to investigate the thermal performance of the PEO/PBAT/HNTs@AlPO_4_ films. Differential scanning calorimetry (DSC) was used to investigate the crystallization behavior of the PEO/PBAT/HNTs@AlPO_4_ films. The rheological analysis was performed to investigate the dynamic rheological performance of the PEO/PBAT/HNTs@AlPO_4_ films.

## 2. Materials and Methods

### 2.1. Materials

PEO (*M*_w_ = 100,000 Da) was purchased from Shanghai Macklin Biochemical Co., Ltd., Shaghai, China. PBAT (*M*_n_ = 20,000 Da) was purchased from Shanxi Jinhui Zhaolong High-Tech Co., Ltd., Shanxi, China. Its mass density is 1.24 g/cm^3^ and its melt flow rate (MFR) at 190 °C is about 5 g/10 min with 2.16 kg. H_3_PO_4_ (≥85%) and Al(OH)_3_ (≥99%) were purchased from Sigma Aldrich. HNTs (Al_2_Si_2_O_5_(OH)_4_·nH_2_O) was obtained from Xi’an Mingchuangda Biotechnology Co., Ltd., Xi’an, China. The HNTs length was 0.2–2 μm, and the inner and outer diameters were 10–40 nm and 40–70 nm, respectively. Silane coupler γ-methacryloxypropyltrimethoxysilane (CG-570, 99.8%) with density of about 1.070 g/cm^3^ and refractive index of around 1.425 at 25 °C was purchased from Nanjing Chengong Silicone Material Co., Ltd., Nanjing, China. Dichloromethane (CH_2_Cl_2_, AR), ethanol (AR), hydrochloric acid (HCl, AR), and NH_3_·H_2_O (AR) were purchased from the Sinopharm Group Chemical Reagent Co., Ltd., Shanghai, China. All chemicals were used as received without any further purification.

### 2.2. Fabrication and Modification of HNTs@AlPO_4_

HNTs@AlPO_4_ was fabricated using a previously reported one-pot heterogeneous precipitation strategy [33]. First, we prepared 160-mL HNT aqueous dispersion (6 wt.%), adjusted the pH of the mixture to 2 using HCl, and labelled this dispersion as A. Second, we prepared a mixed aqueous solution of H_3_PO_4_ and Al(OH)_3_, and labelled this solution as B. For B solution, Al(OH)_3_ was mixed with H_3_PO_4_ in 10 mL deionized water, and stirred to obtain a homogeneous solution at room temperature. Then, we added B dropwise to A, and stirred the mixture at room temperature for 30 min. Afterward, we adjusted the pH of the reaction mixture to 4.8 using NH_3_·H_2_O (27%). Finally, HNTs@AlPO_4_ was obtained after centrifugation, washing, and drying. To improve the interfacial compatibility between inorganic particles and organic polymers, HNTs@AlPO_4_ was modified by a silane coupling agent, CG-570, to obtain organically modified HNTs@AlPO_4_ (o-HNTs@AlPO_4_) using our previously modified method [31]. The detailed procedure is as follows: 0.5 g silane coupling agent CG-570 was added into 60 mL ethanol/H_2_O (*v*/*v* 18:1) and the pH of the mixed solution was adjusted to 4–6 with formic acid. Then, the above CG-570 solution was added into 6.25 g HNTs@AlPO_4_ and stirred at 80 °C for 3 h. The final product was precipitated in ethanol by centrifugation at 5000 rpm for 3 min and dried at 60 °C for 24 h.

Table 1 shows the sample designation (I–IV behand HNTs@AlPO_4_ represents increased Al(OH)_3_/H_3_PO_4_ content) and corresponding content of each component, Figure 1 shows the preparation and modification progress, and all samples were synthesized via the same procedure.

### 2.3. Preparation of PEO/PBAT/o-HNTs@AlPO_4_ Films

The PEO/PBAT/o-HNTs@AlPO_4_ films were prepared using a traditional solution casting method. The detailed procedure is as follows: 2.8-g PEO and 1.2-g PBAT (PEO/PBAT: 70/30) were dissolved in 80 mL CH_2_Cl_2_ and stirred at room temperature until a homogeneous solution was obtained. Then, o-HNTs@AlPO_4_ was added with 1%, 3%, 5%, and 7%, respectively. The o-HNTs@AlPO_4_ was stirred vigorously until the inorganic particles were completely dispersed in the polymeric matrix. Afterward, the film-forming solutions were poured into a polytetrafluoroethylene (PTFE) dish, the solvent was evaporated completely at room temperature for 4 h, and the final dried films were kept in a desiccator at room temperature. Table 2 shows the sample designation and corresponding content of each component, and PEO/PBAT/o-HNTs@AlPO_4_x (x is the percentage of o-HNTs@AlPO_4_) was prepared via the same procedure.

### 2.4. Characterization

An FEI Quanta 250 FEG field-emission SEM (FEI Inc., Valley City, ND, USA) was conducted under high vacuum at a voltage of 20 kV to observe the morphologies of the HNTs@AlPO_4_ nanocomposites and PEO/PBAT films. XRD (D/MAX2500, Rigaku Corporation, Tokyo, Japan) was conducted with Cu Kα radiation (λ = 1.54 Å) over the range of 5°–45° with a scanning rate of 2°/min to investigate the structural characteristics of HNTs@AlPO_4_. FTIR spectroscopy (Nicolet IS50, Thermo Fisher Scientific Inc., Waltham, MA, USA) was conducted to characterize the HNTs and HNTs@AlPO_4_, with a resolution of 4 cm^−1^, range of 400–4000 cm^−1^, and scan number of 32. A TG 219 F3 thermal analyzer (Netzsch Instruments Co., Ltd., Selb, Germany) was conducted in a nitrogen atmosphere at a flow rate of 60 mL/min, and the PEO/PBAT films (approximately 5–10 mg) were heated from 50 to 700 °C at a scanning rate of 20 °C/min to evaluate the thermal stability. DSC 214 (Netzsch Instruments Co., Ltd., Hanau, Germany) was conducted at a heating rate of 10 °C/min to investigate the crystallization of the PEO/PBAT films. Under N_2_ protection, the temperature was increased from 25 °C to 230 °C for 2 min at a ramp rate of 10 °C/min, then lowered to 25 °C for 2 min heated to 230 °C. Model 3365 (Instron Corporation, Boston, MA, USA) was used to determine the tensile performance of the PEO/PBAT films (sample size: 120 × 10 × 0.1 mm). The stretching rate was 5 mm/min. HAAKE MARS III (PolyLab OS., Thermo Fisher Scientific Inc., Waltham, MA, USA) was used to investigate the dynamic rheological properties of the PEO/PBAT films. The size of the film was 20 × 20 × 0.1 mm and the test gap of 1 mm at a temperature of 190 °C. During the test, the strain was 0.5% and the range of frequency was 0.01–100 Hz.

## 3. Results and Discussions

### 3.1. Structural and Morphological Characterization of HNTs@AlPO_4_

In this section, AlPO_4_ was coated on the HNT surfaces to fabricate HNTs@AlPO_4_ using the heterogeneous precipitation method. We investigated the effect of the AlPO_4_ content on the morphology of HNTs@AlPO_4_ to obtain HNTs@AlPO_4_ with a sufficient coating amount and uniform aluminum phosphate dispersion. Figure 2 shows the SEM images of the pure HNTs and HNTs@AlPO_4_ nanocomposites. Figure 2a shows that the pure HNTs had a substantial amount of bulk structure in addition to the tubular structure, and the HNT surfaces were smooth and free of impurities. The HNT surfaces became rough after reacting with AlPO_4_, and the AlPO_4_ nanoparticles were uniformly precipitated on the HNT surfaces (Figure 2b,c). The amount of precipitation gradually increased as the addition of AlPO_4_ content increased, and Figure 2d shows the optimum encapsulation before agglomeration. Figure 2e shows the self-aggregation of the AlPO_4_ nanoparticles due to the limited surface area of the HNTs. In summary, these SEM results showed that AlPO_4_ precipitated on the surface of HNTs in the form of sphere-like nanoparticles and there is an optimal ratio for the reacting raw materials.

The XRD measurements were conducted to further investigate the morphological and structural information of the HNTs@AlPO_4_ composites, and Figure 3a shows the XRD curves of the pure HNTs and HNTs@AlPO_4_ composites. The XRD curves revealed that pure HNTs had peaks at 2θ = 12.06, 19.96, 21.90, and 26.58. A new peak appeared at 2θ = 18.34 after coating with AlPO_4_; as the molarity of AlPO_4_ increased, this peak became more obvious, and HNTs@AlPO_4_III had the highest peak, implying that HNTs@AlPO_4_III was the best sample. The HNTs@AlPO_4_ composite of HNTs is a silicate mineral with the corresponding molecular formula of Al_2_O_3_·2SiO_2_·4H_2_O [34,35]. We selected HNTs@AlPO_4_III for further experiments according to the SEM and XRD results, and still named it after HNTs@AlPO_4_. To improve the compatibility of HNTs@AlPO_4_ with the polymeric matrix, an organic modification was conducted using a silane coupling agent to obtain o-HNTs@AlPO_4_, and Figure 3b shows the IR curves of HNTs, HNTs@AlPO_4_, and o-HNTs@AlPO_4_. The absorption peaks at 3690 and 3620 cm^−1^ were attributed to the stretching vibrations of –OH in the HNTs, and the absorption bands in the range of 950–1093 cm^−1^ were attributed to the stretching vibrations of the Si–O groups in HNTs [36,37]. The absorption peaks at 2960 and 2855 cm^−1^ were attributed to the stretching vibrations of –CH_3_ and –CH_2_– in the silane coupling agent, CG-570, after organic modification.

### 3.2. Thermal Properties of PEO/PBAT/o-HNTs@AlPO_4_ Films

The TGA and DSC measurements are widely used to investigate the thermal stability and crystallization behavior of various materials, respectively. Figure 4 shows the TGA and DSC curves of the PEO/PBAT and PEO/PBAT/o-HNTs@AlPO_4_ films, and Table 3 lists the corresponding data. The TG and DTG curves showed that the samples had only one weight-loss stage, and their thermal degradation behavior mainly occurred between 350 °C and 450 °C. The corresponding data revealed that the initial decomposition temperature (*T*_0_) of PEO/PBAT was 307.59 °C, and as the o-HNTs@AlPO_4_ content increased, *T*_0_ increased first and then decreased, and the *T*_0_ of all of the PEO/PBAT/o-HNTs@AlPO_4_ films were higher than those of PEO/PBAT. Moreover, the carbon residue of PEO/PBAT was 6.18 wt.%, and as the o-HNTs@AlPO_4_ content increased, the carbon residue of the PEO/PBAT/o-HNTs@AlPO_4_ film became 9.45, 10.18, 11.80, and 13.07 wt.%, with only a minor change in the maximum decomposition temperature (*T*_max_). In summary, adding the appropriate o-HNTs@AlPO_4_ content could improve the thermal stability of the PEO/PBAT blends. The DSC results show that the melting temperature (*T*_m_) and crystallization temperature (*T*_c_) of PEO/PBAT were 61.29 °C and 36.24 °C, respectively; the addition of o-HNTs@AlPO_4_ had little effect and only marginally increased the *T*_m_ and *T*_c_. When the o-HNTs@AlPO_4_ content was 7 wt.%, the *T*_c_ suddenly decreased to 34.32 °C. This phenomenon may be due to excess o-HNTs@AlPO_4_ in PEO/PBAT/o-HNTs@AlPO_4_ 7, making o-HNTs@AlPO_4_ difficult to disperse and resulting in localized agglomeration. Thus, the corresponding thermal stability and crystallization performance decline. *ΔH*_m_ in Table 3 represents the enthalpy of the melting process, and the corresponding values of PEO/PBAT/o-HNTs@AlPO_4_ films are listed. In conclusion, the TG and DSC results revealed that the o-HNTs@AlPO_4_ particles are compatible with PEO/PBAT, that the addition of o-HNTs@AlPO_4_ to polymeric blends increases the stability and crystallinity, and that there is an optimum o-HNTs@AlPO_4_ content.

### 3.3. Mechanical Properties of PEO/PBAT/o-HNTs@AlPO_4_ Films

Model 3365 (Instron Corporation, Boston, USA) was used to conduct the tensile performance measurements to investigate the effects of o-HNTs@AlPO_4_ on the mechanical properties of PEO/PBAT, and Table 4 lists the results of the tensile strength (MPa), flexural strength (MPa), and elongation at break (%) for the PEO/PBAT films with various o-HNTs@AlPO_4_ contents. The tensile strength, flexural strength, and elongation at break of pure PEO/PBAT films were 3.59 ± 0.08 MPa, 202.72 ± 6.81 MPa, and 49.23 ± 3.85%, respectively. When o-HNTs@AlPO_4_ was added to the blend, the strength and elongation both increased first and then decreased, and a certain o-HNTs@AlPO_4_ content could simultaneously improve the strength and toughness of the PEO/PBAT films. The tensile strength of the PEO/PBAT films improved by 25.91% when 3 wt.% o-HNTs@AlPO_4_ was added. The flexural strength and elongation at break were improved by 24.56% and 44.87%, respectively, when 5 wt.% o-HNTs@AlPO_4_ was added. However, the strength and toughness were simultaneously reduced when 7 wt.% o-HNTs@AlPO_4_ was added. In summary, the o-HNTs@AlPO_4_ particles are compatible with the PEO/PBAT blend and can improve the interfacial compatibility of PEO and PBAT. Moreover, excessive o-HNTs@AlPO_4_ particles prevent dispersion in the polymeric matrix, lowering the mechanical properties of the films [38]. This mechanical property result is consistent with the above TG and DSC results.

### 3.4. Cryofractured Surfaces of PEO/PBAT/o-HNTs@AlPO_4_ Films

Figure 5 shows the SEM results of the cryofractured surfaces of the PEO/PBAT blends with various o-HNTs@AlPO_4_ contents, and the inserted white dotted ellipses represent PBAT, and the inserted red dotted rectangles represent o-HNTs@AlPO_4_. The PEO/PBAT blends mainly displayed a matrix–dispersed phase morphology with PEO continuous and irregular or spherical PBAT-dispersed phases, and the short axis of the dispersed phase was almost less than 1 μm.

Figure 5a shows that the cross-section of pure PEO/PBAT was relatively rough with several ultrafine particles, PBAT was mainly dispersed in an irregular shape in the continuous phase, and there was an obvious phase separation at the two-phase interface. When o-HNTs@AlPO_4_ was added to PEO/PBAT, the irregular PBAT decreased and some spherical PBAT appeared, and the cross-section of PEO/PBAT/o-HNTs@AlPO_4_ became smooth (Figure 5b–d). Furthermore, the spherical-dispersed phase gradually merged with the continuous phase, and the rod-like o-HNTs@AlPO_4_ was primarily observed in the PEO continuous phase, with an almost aligned orientation. This arrangement may be due to the rapid volatilization of CH_2_Cl_2_ during the solution pouring process as well as the uniform distribution of o-HNTs@AlPO_4_ in the PEO matrix. The cryofractured surfaces became smoother and denser as the o-HNTs@AlPO_4_ content increased, and the spherical PBAT-dispersed phase gradually merged with the PEO continuous phase, resulting in the spherical structure decreasing or even disappearing in Figure 5c,d. Figure 5e shows the self-aggregation of o-HNTs@AlPO_4_, and the macroscopic phase separation between the inorganic particles and the polymeric matrix occurred due to the excessive addition of o-HNTs@AlPO_4_. In summary, the appropriate o-HNTs@AlPO_4_ content has good compatibility with the PEO/PBAT blend, and could further reduce the surface tension and improve the compatibility between PEO and PBAT, where o-HNTs@AlPO_4_ may serve as a compatibilizer in this blending system.

### 3.5. Rheological Properties of PEO/PBAT/o-HNTs@AlPO_4_ Films

The rheological behavior of polymeric blends is not only related to their microstructure and processability, but also plays a vital role in their interfacial interactions [39,40]. In this study, the localization of o-HNTs@AlPO_4_ particles with rough surfaces was observed to have good interfacial compatibility with PEO and PBAT. Figure 6 shows the effect of o-HNTs@AlPO_4_ on the storage and loss moduli of the PEO/PBAT blends. PEO/PBAT/o-HNTs@AlPO_4_ had a higher storage and loss moduli in both the low- and high-frequency regions than PEO/PBAT, demonstrating the melt reinforcement effect of inorganic particles. Moreover, with an increase in the o-HNTs@AlPO_4_ content, both the storage and loss moduli first increased and then decreased, and PEO/PBAT/o-HNTs@AlPO_4_5 reached the maximum. The changes in G’ were more than G”, indicating that the elasticity was greater than the viscosity of PEO/PBAT/o-HNTs@AlPO_4_. These results revealed that with the increase in frequency, the viscoelastic behavior became better for all samples, and after the addition of o-HNTs@AlPO_4_, the viscoelastic behavior of PEO/PBAT/o-HNTs@AlPO_4_ were even better than that of the pure PEO/PBAT. It may be that the excessive content of o-HNTs@AlPO_4_ reduces the cross-linking reaction between PBAT and PEO and enhances the disentanglement ability between molecular chains. Inorganic particles with good dispersibility could improve the rheological properties of the polymer blends [41,42]. In this study, o-HNTs@AlPO_4_ was dispersed in the PEO/PBAT matrix in an oriented arrangement, increasing the interfacial area due to the good compatibility between o-HNTs@AlPO_4_ and PEO and PBAT, and this result is similar to the trend in the thermal and mechanical properties discussed above.

Figure 7 shows the relationship between the complex viscosity and angular frequency of the PEO/PBAT blends. The addition of o-HNTs@AlPO_4_ can increase the complex viscosity in all frequency regions compared with pure PEO/PBAT blends. PEO/PBAT/o-HNTs@AlPO_4_5 also reached the maximum, and the excessive addition of o-HNTs@AlPO_4_ resulted in decreased compatibility and macroscopic phase separation. Furthermore, the complex viscosity curves of the PEO/PBAT/o-HNTs@AlPO_4_ films merged into one in the high-frequency region (~100 rad/s) because the higher shear rate destroyed the interfacial interaction force between o-HNTs@AlPO_4_, PEO, and PBAT.

## 4. Conclusions

In summary, aluminum phosphate-coated halloysite nanotubes (HNTs@AlPO_4_) were fabricated using a modified one-pot heterogeneous precipitation method, and the influence of various AlPO_4_ contents on the morphology of HNTs@AlPO_4_ was investigated. Furthermore, organically modified HNTs@AlPO_4_ (o-HNTs@AlPO_4_) was introduced into the PEO/PBAT blends, and the influence of o-HNTs@AlPO_4_ on the morphological, mechanical, and rheological properties of the PEO/PBAT/o-HNTs@AlPO_4_ films was systematically investigated. Our results indicate that o-HNTs@AlPO_4_ fillers act as a compatibilizer, improving not only the mechanical properties but also the rheological properties of PEO/PBAT. This study provides a new route for mechanically well-balanced PEO blends and promotes the application of PEO films.

## Figures and Tables

**Figure 1 nanomaterials-12-02896-f001:**
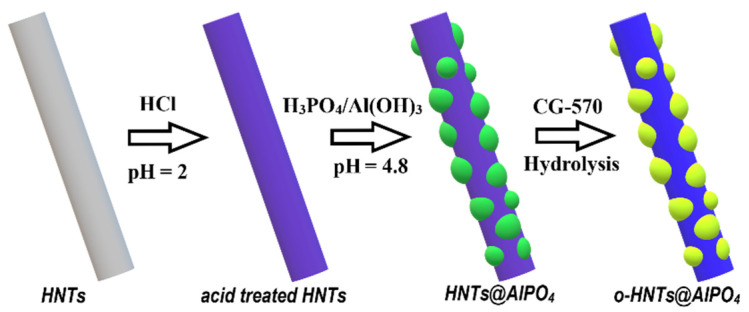
A schematic diagram of the preparation and modification of HNT_S_@AlPO_4_.

**Figure 2 nanomaterials-12-02896-f002:**
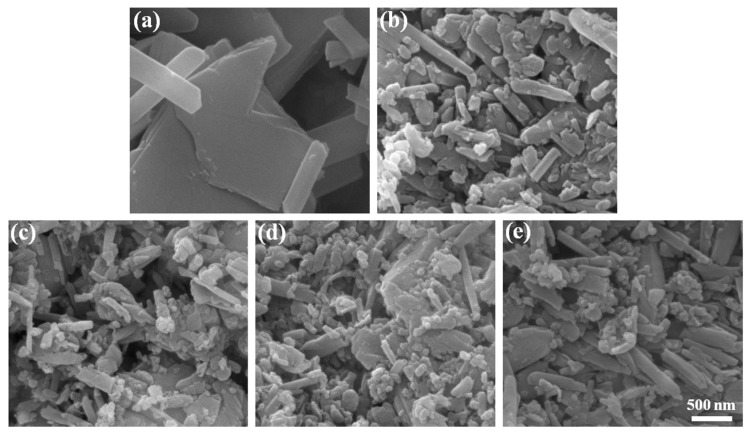
The SEM images of the pure HNTs and the HNTs@AlPO_4_ nanocomposites as synthesized. (**a**) HNTs, (**b**) HNTs@AlPO_4_I, (**c**) HNTs@AlPO_4_II, (**d**) HNTs@AlPO_4_III, and (**e**) HNTs@AlPO_4_IV. The scale bar in the last image can be applied to the other images.

**Figure 3 nanomaterials-12-02896-f003:**
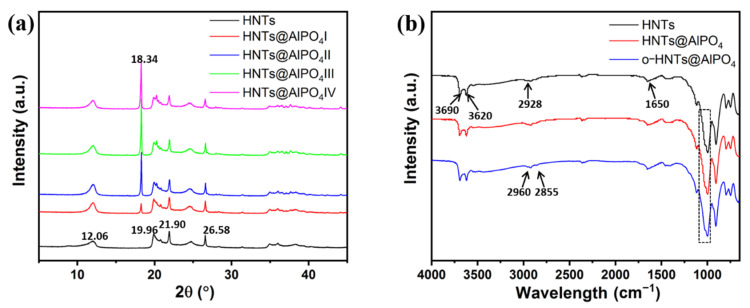
(**a**) The XRD curves of the pure HNTs and HNTs@AlPO_4_ composites and (**b**) IR curves of HNTs, HNTs@AlPO_4_, and o-HNTs@AlPO_4_.

**Figure 4 nanomaterials-12-02896-f004:**
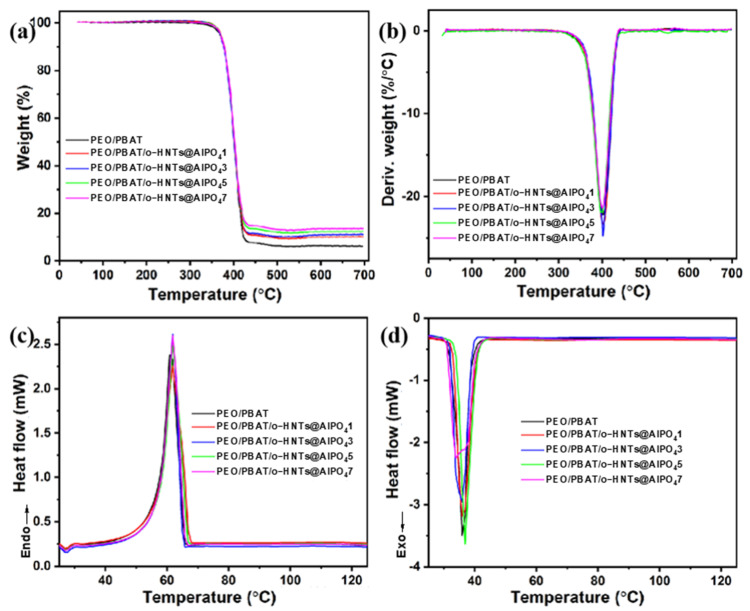
(**a**) The TG, (**b**) DTG, and (**c**,**d**) DSC curves of the PEO/PBAT and PEO/PBAT/o-HNTs@AlPO_4_ films, where (**c**,**d**) represent the heating and cooling processes, respectively.

**Figure 5 nanomaterials-12-02896-f005:**
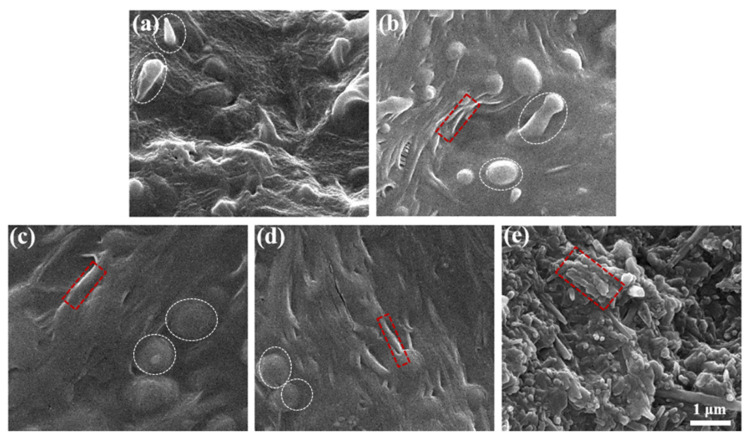
The SEM images of the cross-sections of the PEO/PBAT and PEO/PBAT/o-HNTs@AlPO_4_ films. (**a**) PEO/PBAT, (**b**) PEO/PBAT/o-HNTs@AlPO_4_1, (**c**) PEO/PBAT/o-HNTs@AlPO_4_3, (**d**) PEO/PBAT/o-HNTs@AlPO_4_5, and (**e**) PEO/PBAT/o-HNTs@AlPO_4_7. The scale bar in the last image can be applied to the other images, and the inserted white dotted ellipses represent PBAT, and the inserted red dotted rectangles represent o-HNTs@AlPO_4_.

**Figure 6 nanomaterials-12-02896-f006:**
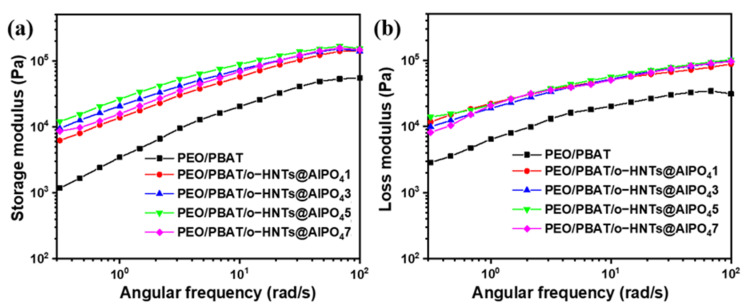
(**a**) The storage modulus and (**b**) loss modulus versus angular frequency for the PEO/PBAT and PEO/PBAT/o-HNTs@AlPO_4_ films at 190 °C.

**Figure 7 nanomaterials-12-02896-f007:**
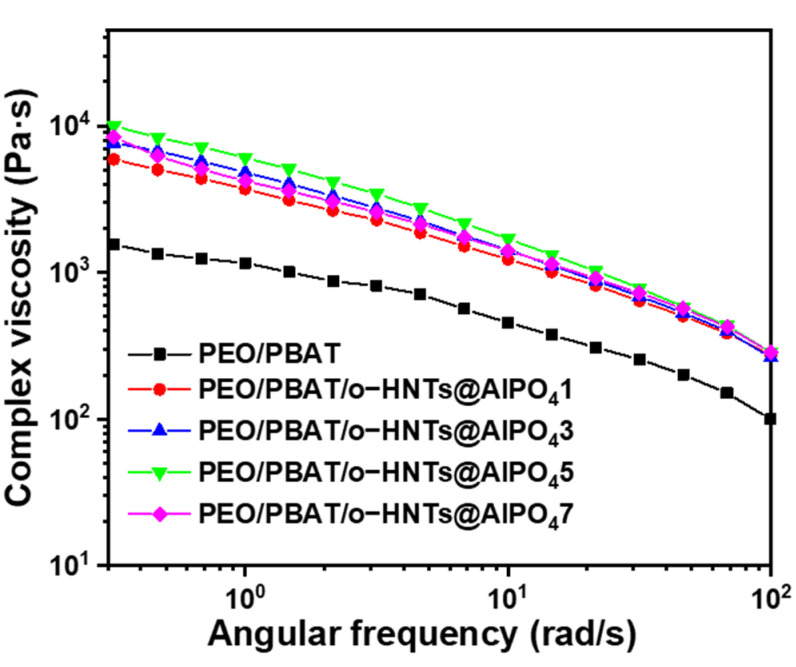
Plots of the complex viscosity versus angular frequency for the PEO/PBAT and PEO/PBAT/o-HNTs@AlPO_4_ films at 190 °C.

**Table 1 nanomaterials-12-02896-t001:** The details of the sample designations for the HNTs@AlPO_4_ nanocomposites.

Sample Designation	HNTs (g)	Al(OH)_3_ (g)	H_3_PO_4_ (g)
HNTs@AlPO_4_I	10	0.58	4.33
HNTs@AlPO_4_II	10	1.17	8.67
HNTs@AlPO_4_III	10	1.75	13.00
HNTs@AlPO_4_IV	10	2.33	17.33

**Table 2 nanomaterials-12-02896-t002:** Formulations of the PEO/PBAT/o-HNTs@AlPO_4_ films.

Sample Designation	PEO (g)	PBAT (g)	HNTs@AlPO_4_ (g)
PEO/PBAT	2.8	1.2	0
PEO/PBAT/o-HNTs@AlPO_4_1	2.8	1.2	0.04
PEO/PBAT/o-HNTs@AlPO_4_3	2.8	1.2	0.12
PEO/PBAT/o-HNTs@AlPO_4_5	2.8	1.2	0.20
PEO/PBAT/o-HNTs@AlPO_4_7	2.8	1.2	0.28

**Table 3 nanomaterials-12-02896-t003:** Data of the TG and DSC of the PEO/PBAT/o-HNTs@AlPO_4_ films.

Sample Designation	*T*_0 _ (°C)	*T*_max_ (°C)	Carbon Residue (wt.%)	*T*_m_ (°C)	*T*_c _ (°C)	*ΔH_m_* (J/g)
PEO/PBAT	307.59	401.40	6.18	61.29	36.24	86.62
PEO/PBAT/o-HNTs@AlPO_4_1	309.09	401.67	9.45	61.72	36.45	88.74
PEO/PBAT/o-HNTs@AlPO_4_3	310.59	401.78	10.18	61.89	36.52	86.20
PEO/PBAT/o-HNTs@AlPO_4_5	309.83	401.86	11.80	61.98	36.98	88.44
PEO/PBAT/o-HNTs@AlPO_4_7	308.42	401.37	13.07	61.97	34.32	89.07

**Table 4 nanomaterials-12-02896-t004:** The mechanical properties of the PEO/PBAT/o-HNTs@AlPO_4_ films.

Sample Designation	Tensile Strength (MPa)	Flexural Strength (MPa)	Elongation at Break (%)
PEO/PBAT	3.59 ± 0.08	202.72 ± 6.81	49.23 ± 3.85
PEO/PBAT/o-HNTs@AlPO_4_1	3.82 ± 0.12	238.82 ± 7.35	54.32 ± 3.61
PEO/PBAT/o-HNTs@AlPO_4_3	4.52 ± 0.09	241.03 ± 8.18	66.20 ± 4.23
PEO/PBAT/o-HNTs@AlPO_4_5	3.68 ± 0.15	252.51 ± 7.64	71.32 ± 4.57
PEO/PBAT/o-HNTs@AlPO_4_7	2.52 ± 0.21	187.57 ± 8.56	47.22 ± 4.73

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
