# Peer review of "Synthesis of Aluminum Phosphate-Coated Halloysite Nanotubes: Effects on Morphological, Mechanical, and Rheological Properties of PEO/PBAT Blends"

_nanomaterials, 2022, doi:10.3390/nano12172896_

Round 1

Reviewer 1 Report

All acronyms used in the manuscript have to be explained when used for the first time: HNTs, HNTs@AlPO4,  PEO, PBAT,

It is well known that an addition of a different nanofiller can significantly improve mechanical as well as thermal properties. In my opinion, however,  it is unjustified to claim that they can play the role of compatibilizers. Please provide the compatibilizer definition and explain the mechanism of the compatibilization process by introduction of montmorillonite, halloysite nanotubes (HNTs), SiO2, and Al2O3 and studied HNTs@AlPO4 in order to support the thesis.

Please clearly present the novelty and the aim of the work in the “Introduction” section.

In the “Materials” section please provide more information concerning used polymers.

Explain why the pH changed during the fabrication of  HNTs@AlPO4

The Authors claim that “First, prepare 160-mL HNT aqueous solution (6 wt.%)” – To my knowledge HNT cannot be dissolved in water.

Please provide the scheme of HNTs@AlPO4 fabrication and modification.

Section 2.3: The Authors indicate that “o-HNTs@AlPO4 was added and stirred vigorously until the inorganic particles were completely dispersed in the polymeric matrix”. Please explain how the eventual dispersion of particles was established.

The size of polystyrene Petri dishes and the exact volume of Ch2Cl2 used for dissolving polymers as well as the time of solvent evaporation should be indicated.

“In this section, AlPO4 was coated on the HNT surfaces to fabricate HNTs@AlPO4 using the heterogeneous precipitation method. We investigated the effect of AlPO4 content on the morphology of HNTs@AlPO4 to obtain HNTs@AlPO4 with sufficient coating amount and uniform aluminum phosphate dispersion.” – The quoted part of the manuscript is unclear and should be rephrased.

Figure 1. The scale has to be indicated on all presented figures (a-e) and the size of HNT has to be marked.

“Figure 1d shows the optimum encapsulation before agglomeration” Please clarify this statement.

A new peak was ADDED” (XRD section). Please explain this statement.

What is the ratio between PEO and how was PBAT established? 

The structural study of the obtained PEO-PBAT-based composites has to be provided.

In summary, HNTs@AlPO4 particles are compatible with PEO/PBAT blend and can improve the interfacial compatibility of PEO and PBAT” Please clarify this statement and clearly explain how is it possible that after the introduction of the filler, the elongation at break increased.

(section 3.4) In an aim to establish the dispersion of filler in the polymer matrix the cross-section that shows all samples has to be added. In the present form, the analysis of dispersion is impossible. Moreover, particular components have to be marked in SEM images.

Reviewer 2 Report

The paper titled "Synthesis of Aluminum Phosphate-Coated Halloysite Nano-tubes: Effects on Morphological, Mechanical, and Rheological Properties of PEO/PBAT Blends" deals with the preparation of coated halloysite, its modification with silanes and its use for the production of nanocomposite films using PEO and PBAT.

Unfortunately, in my opinion the paper has too many flaws to allow publication in Nanomaterials, here my main points of criticism

1) The overall presentation of how the experiments were performed is poor, and in particular:

2.1 Materials

PEO molecular weight: is it Mn or Mw?

PBAT injection molding grade: which exact grade? no information is given. MFI? Other data? Why using injection molding grade to obtain films?

Silane Coupler: it is unconceivable to deal with a reagent without indicating its chemical composition. Which silane is it?

HTN: where does it come from? data? average dimensions? other data? it is not mentioned.

2.2

When preparing a mixed solution of H3PO4 and Al(OH)3 no indications are given. Quantities, ratios etc? This is a new paper, authors must indicate how they prepare it, not just refer to other previous papers, as they did for the HNT aqueous solution.

The same holds true for silane addition.

2.3

The nanocomposite obtained is not "synthesised": no synthesis is involved, they just blended two polymers with HNT. It is not a synthesis. "A certain amount of CH2Cl2" is unconceivable for a paper? How much CH2Cl2? I cannot replicate the experiment is quantities are not given.

I seriously doubt that authors used polystyrene petri, since polystyrene dissolves immediately in CH2Cl2

In 2.2 authors state that they prepared 4 HNTs@AlPO4 but in Table 2 they do not indicate which one they used for the preparation of films. 1-3-5-7 relate to the concentration of HTNs@AlPO4 in the blend. 

3.1

In SEM: EDS would be useful to understand if AlPO4 distributes efficiently on HTN

Why, checking XRD, the highest peak at 18.34 means that it is the "best sample"? This is not explained.

Why is the highest peak with III and not with IV?

How is XRD after silane addition?

3.2.

This is probably the weakest part of the paper. Authors cannot discuss TGA and DSC in such a poor way.

T0 (authors do not state how they determine it) changes from 307.59°C (a precision to 0.01°C is really too much) to 310.59°C: this is really a minor change and it is not significant. Authors should simply state that T0 is almost unchanged and only slightly increased. I repeat that 3°C increase is really just a minor increase.

Authors state that adding HTN increase the thermal stability after assessing the residual weight: it is quite obvious, since HTN is an inorganic and, I repeat, a T0 increased by 3°C does not justifiy such a statement.

Melting temperature: authors state that Tm of the blend is 61°C. This is a mistake: authors themselves state that the blend is heterogeneous and should consider that the melting point of PEO 10000 is about 60°C, therefore they are only seeing the melting of PEO. PBAT melts at about 125-140°C according to the specific PBAT, but authors stop the DSC before.

Again, when dealing with Tc, authors state that a 2°C difference is significant and, most important, correlate the decrease in Tc with the lower T0 in TGA. This is completely wrong.

3.5

Little/no difference is visible in Rheological curves when HNT is used. I would expect higher differences between HTN1 and 7, for example, being the latter a 7-fold increase in the inorganic fraction present in the blend.

Moreover, in 2.4, no indication is given about how the tests were performed. PEO and PBAT are highly hygroscopic: did authors dried the polymers? measuring system? gap? temperatures?

Other concerns

Wording in "Introduction":

"refractory" polymeric films...why refractory ? very bad wording

"biopolymers" are polymers produced by nature, authors cannot confuse bioplastics with biopolymers

PBAT is not rubber-like: rubbers are very different, maybe authors mean "elastomer" but I do not know

Many other bad wording examples are present in the paper.

I suggest authors to completely revise their paper and be more accurate in the experiments description. Moreover authors should not magnify the results obtained and be careful on the real improvements observed

Reviewer 3 Report

The paper discusses the effects on morphological, mechanical and rheological properties of PEO/PBAT blends of aluminum phosphate-coated halloysite nanotubes. The paper is quite interesting because demonstrates the reinforcing effect of the nanotubes accompanied by the compatibilizing effect of the coating.

Some points, however, must be clarified.

1.     Introduction “Polymeric films have been widely used in heavy and light industries, dentistry, petrochemical …” The more important use of polymeric films is packaging! Where polymeric films are used in dentistry or petrochemical?  What means polymeric films in your paper?

2.     All the nanofillers used in this paper are named HNTs@AIPO4, but at pag. 2 the organically modified H NTs@AIPOare named (o-HNTs@AIPO4). This denomination is used only at pag. 3 but not in the tables and figures. The authors should revise the denominations of the samples.

3.     The tables 4 and 5 and the Figures 4-6 reports data measured as a function of the amount of nanofiller. Which nanofiller sample is used (I, II, III, IV ?)

4.     Fig. 5 the slope of G’ vs frequency is an important evidence of the viscoelastic behavior of the polymeric systems. The authors should discuss the change of slope of the G’ curves in the presence of the nanofiller.

Round 2

Reviewer 1 Report

I have to admit that the Authors have significantly improved the submitted manuscript.

Author Response

Thank you for your help and encourage of our manuscript

Reviewer 2 Report

I thank the authors for the many changes made on the paper but still I do not find the paper suitable for publication.

Here some main concerns: 

1) According to authors Mw of PEO is about 100000: no measuring unit is given… I suppose it’s Daltons, but it is disappointing that authors writing a paper on blend preparation do not show knowledge of the simplest ways to indicate polymers molecular weights.

2) PBAT Ecoworld has a declared MFI of 3-5 g/10’ at 190°C under a 2.16 Kg load: why authors are so generic ( 5 g/10’) and indicate 2Kgs and not 2.16 that is the weight indicated also by ISO and ASTM rules? Authors should not indicate to me where other information can be provided, but should put in the paper the ones useful to make other researchers able to reply their experiments, also using different PBAT trademarks having the same properties. This is the basis for the share of knowledge of research papers.

3) I do not really understand the rationale behind the choice of CG-570, now that I see its formula: it is an acrylic silane, therefore it is useful especially when double bonds can react with something. Why did authors use is and not, for example, an epoxy silane, that could react with PBAT and/of PEO? This is a very weak point and this choice makes no sense to me: CG-570 cannot react with the polymer and consequently has a very limited compatibilizing effect. Authors should explain what they expect from this silane

4) In my previous report “When preparing a mixed solution of H3PO4 and Al(OH)3 no indications are given. Quantities, ratios etc? This is a new paper, authors must indicate how they prepare it, not just refer to other previous papers, as they did for the HNT aqueous solution.”

Now quantities of Al(OH)3 and H3PO4 are given in Table 1 but the quantity of water is not indicated

5) In relation to silane, it is a new paper: authors must indicate how to perform silane addition. They can refer to their previous paper but they should anyway describe the procedure.

Moreover, authors call an Ethanol/water 18/1 solution an “aqueous solution”?

Again, I do not find the quantity of silane used, authors in the reply to my comment state “aqueous solution with 8 wt% content relative to acidified HNTs@AlPO4” but a line should be added in table I or somewhere else indicating the quantity of silane used.

6) Now in Figure 1 authors show the preparation: I do not understand the first passage. If at pH=2 HNT dissolves in water, it means that it dissociates somehow, otherwise it would not be soluble. How does the mineral dissociate? what happens then, after reprecipitation? And I do not understand how the silane should react with the coated surface of HNT

7) Again, in the text I cannot see where authors indicate that they use HNTs@AlPO4IV of Table 1

8) Why, if according to authors XRD shows that HNTs@AlPO4III is the best sample, authors use HNTs@AlPO4IV?

9) Rheology in 2.4 “The size of the film is 20 mm × 20 mm × 0.1 mm. The three layers are overlapped at a temperature of 190°C with a gap of 1 mm. During the test, the strain is 0.5% and the range of frequency is 0.01-100 Hz.”

I really do not understand: the gap is 1 mm but the film is 0.1 mm thick?

Authors later state that they defined that samples were dried before the analyses (as an answer to another point I raised) but it seems to me that no trace of this is present in the text.

10) The thermal part has been slightly improved but it is very weak anyway:

“In summary, adding appropriate o-HNTs@AlPO4 content could improve the thermal stability of PEO/PBAT blends”

As stated before, I do not really see such improvements: 2-3°C of difference or  a slightly higher residual weight when higher quantities of o-HNTs@AlPO4 are used.

If PEO/PBAT residue is about 6% and, after adding 7% of o-HNTs@AlPO4, it reaches about 13% I would say that it is highly expected.

If I add 20% of talc it reaches 26%, but it is not higher stability, it is higher inorganic content.

11) According to the new data, the only melting of the blend is at about 60°C. As authors can understand , a 60°C melting makes the blend unsuitable for most common applications. Morevoer, if the peak of PBAT is not visible, it means that somehow the polymer is miscible, compatible with PEO. Have authors checked melting hentalpies comparing them to the ones of standard PEO to assess what changes?

As I stated previously, I am sorry but the paper is still missing many parts and I do not see such big new findings to justify publication.

Reviewer 3 Report

The answers to my comments are well addressed but not the last comment.

In the revised version the authors write:

1. “Compared with the blends of adding o-HNTs@AlPO4, the slope of pure-PEO/PBAT is large at low frequency regions in Figure 6a, and the pure PEO/PBAT tends to be elastic.” 

The elasticity increases with decreasing the slope at low frequency. Indeed, a material is elastic if G’ does not depend on the frequency. The authors must eliminate this mistake and should interpret correctly nthe experimental data.

2. “These results revealed that with the increase of frequency, the viscoelastic became better for all samples, and after the addition of o-HNTs@AlPO4, the viscoelastic of PEO/PBAT/o-HNTs@AlPO4 were better than pure PEO/PBAT. “  

What means "viscoelastic became better"? Do you mean viscoelastic behavior?  And why better?

Author Response

1. “Compared with the blends of adding o-HNTs@AlPO4, the slope of pure-PEO/PBAT is large at low frequency regions in Figure 6a, and the pure PEO/PBAT tends to be elastic.” The elasticity increases with decreasing the slope at low frequency. Indeed, a material is elastic if G’ does not depend on the frequency. The authors must eliminate this mistake and should interpret correctly nthe experimental data.

[Response] Thanks for the helpful comments. We have made revisions in the revised manuscript.

2. “These results revealed that with the increase of frequency, the viscoelastic became better for all samples, and after the addition of o-HNTs@AlPO4, the viscoelastic of PEO/PBAT/o-HNTs@AlPO4 were better than pure PEO/PBAT. “  What means "viscoelastic became better"? Do you mean viscoelastic behavior?  And why better?

[Response] Thanks for the helpful comments. The viscoelastic behavior became better. It may be that the excessive content of o-HNTs@AlPO4 reduces the cross-linking reaction between PBAT and PEO and enhances the disentanglement ability between molecular chains.

Round 3

Reviewer 3 Report

the paper is accepted

Author Response

(The authors gave the same response as above.)
